# Complete Genome Sequencing and Comparative Phylogenomics of Nine African Swine Fever Virus (ASFV) Isolates of the Virulent East African *p72* Genotype IX without Viral Sequence Enrichment

**DOI:** 10.3390/v16091466

**Published:** 2024-09-14

**Authors:** Jean-Baka Domelevo Entfellner, Edward Abworo Okoth, Cynthia Kavulani Onzere, Chris Upton, Emma Peter Njau, Dirk Höper, Sonal P. Henson, Samuel O. Oyola, Edwina Bochere, Eunice M. Machuka, Richard P. Bishop

**Affiliations:** 1International Livestock Research Institute, Nairobi P.O. Box 30709-00100, Kenya; e.okoth@cgiar.org (E.A.O.); sonal.henson@nottingham.ac.uk (S.P.H.); s.oyola@cgiar.org (S.O.O.); mamaingado@gmail.com (E.B.); e.machuka@cgiar.org (E.M.M.); 2Department of Veterinary Microbiology and Pathology, Washington State University, Pullman, WA 99164-1067, USA; cynthia.onzere@wsu.edu (C.K.O.); bishop5030@gmail.com (R.P.B.); 3Viral Bioinformatics Research Centre, University of Victoria, Victoria, BC V8P 5C2, Canada; cupton.uvic@gmail.com; 4Department of Microbiology, Parasitology and Biotechnology, Sokoine University of Agriculture, Morogoro P.O. Box 3019, Tanzania; emmapeter@sua.ac.tz; 5Friedrich-Loeffler-Institut, 17493 Greifswald-Insel Riems, Germany; dirk.hoeper@fli.de

**Keywords:** African swine fever virus, ASFV, p72 Genotype IX, East Africa, field isolates, phylogenomics, rapid whole-genome sequencing

## Abstract

African swine fever virus (ASFV) is endemic to African wild pigs (*Phacochoerus* and *Potamochoerus*), in which viral infection is asymptomatic, and *Ornithodoros* soft ticks. However, ASFV causes a lethal disease in Eurasian domestic pigs (*Sus scrofa*). While Sub-Saharan Africa is believed to be the original home of ASFV, publicly available whole-genome ASFV sequences show a strong bias towards p72 Genotypes I and II, which are responsible for domestic pig pandemics outside Africa. To reduce this bias, we hereby describe nine novel East African complete genomes in p72 Genotype IX and present the phylogenetic analysis of all 16 available Genotype IX genomes compared with other ASFV p72 clades. We also document genome-level differences between one specific novel Genotype IX genome sequence (KE/2013/Busia.3) and a wild boar cell-passaged derivative. The Genotype IX genomes clustered with the five available Genotype X genomes. By contrast, Genotype IX and X genomes were strongly phylogenetically differentiated from all other ASFV genomes. The p72 gene region, on which the p72-based virus detection primers are derived, contains consistent SNPs in Genotype IX, potentially resulting in reduced sensitivity of detection. In addition to the abovementioned cell-adapted variant, eight novel ASFV Genotype IX genomes were determined: five from viruses passaged once in primary porcine peripheral blood monocytes and three generated from DNA isolated directly from field-sampled kidney tissues. Based on this methodological simplification, genome sequencing of ASFV field isolates should become increasingly routine and result in a rapid expansion of knowledge pertaining to the diversity of African ASFV at the whole-genome level.

## 1. Introduction

African swine fever is rapidly lethal in naive domestic pigs and Eurasian wild boar [1]. It is caused by a large DNA virus, with a genome 170–194 kilobases in size, which is the best characterized member of the Asfarviridae [2]. The virus infects porcine macrophages, and there is a wildlife reservoir in African wild pigs, warthogs (*Phacochoerus*) and bushpigs (*Potamochoerus*), in which the virus does not induce clinical disease [3]. The virus is spread between warthogs by soft ticks in the genus *Ornithodoros* that inhabit warthog burrows—a route of transmission which was important historically for the infection of domestic pigs [3]. The main route of transmission, even in the endemic areas of Africa, is now believed to be a direct one between domestic pigs [1].

The disease caused by ASFV recently re-emerged as a threat to the global pig industry as a result of transmission of the virus from Southeastern Africa to the Caucasus [4,5] and subsequently Russia and Eastern Europe, with major economic consequences [6]. The spread of the virus to China, where an estimated 450 million domestic pigs are reared, many in small-scale systems, combined with subsequent further transmission to Southeast Asia, has created a pig pandemic with severe consequences for food security in East Asia and globally [7].

The complete genome sequences currently available for ASFV still constitute a minority of the genotypes defined by sequencing of the 3′ end of the gene encoding the viral p72 major surface protein [8], with a very strong predominance of sequences derived from the index Genotype II virus that is the cause of the ongoing porcine pandemic [9]. Initially, the majority of complete ASFV genomes were derived from the West African Genotype I that devastated the pig industry in Portugal and Spain between 1958 and 1997 [1] and was historically the main target of molecular research on ASFV. Since the introduction of a p72 Genotype II ASFV to Georgia from Southeastern Africa or Madagascar [4,5], the focus has switched to the production of multiple genome sequences ultimately derived from a Genotype II index isolate introduced to Georgia, which was responsible for the ongoing Eurasian ASFV pig pandemic [9]. Recently, the first indigenous African Genotype II virus from Rukwa district, Southern Tanzania, was published [10,11]. ASFV genomes from Africa (including the virulent East African Genotype IX [12,13]) represent a significant potential future global threat to the global pig industry, should they be exported from the continent. The fact that a Genotype IX virus has caused disease in pig farms in the suburbs of the major port city of Mombasa [14] clearly indicates that the risk of a further spread is real. However, African ASFV genomes have been neglected in terms of whole-genome sequencing and analyses, and this study represents a contribution towards filling the gap in global ASFV whole-genome data.

Initial approaches to complete genome sequencing of ASFV using next-generation platforms initially involved either in vivo infection of pigs, followed by labor-intensive purification of the virus on sucrose gradients [12], or long-range PCR approaches that are limited by the requirement for conserved primers [15]. Recently, viral DNA enrichment through the removal of methylated vertebrate sequences from the domestic pig host was used to successfully to determine the genomes of five Genotype IX viral isolates from Eastern Uganda [16].

Herein, we report the determination of eight genomes from primary virus isolates, and one lab-adapted isolate from Kenya and Uganda, isolated from both clinically symptomatic and asymptomatic domestic pigs, and compare them phylogenetically with selected well-annotated ASFV genomes. The novel genomes were all classified as p72 Genotype IX. They were generated either from DNA directly extracted from tissue from infected pigs sampled in the field, or from primary macrophage culture isolates. However, neither protocol involved the enrichment of viral nucleic acids, through either experimental in vivo infection of pigs, or viral DNA in vitro. These genomes represent the largest ASFV population genomic dataset produced from any endemic area in Africa to date, and strongly indicate that recent ASFV outbreaks spanning a region from eastern Uganda to central Kenya isolated during the period between 2006 to 2013 were caused primarily by ASFV Genotype IX and are generally very similar at the whole-genome level. The simplified procedure that we describe should accelerate the process of determining complete ASFV genomes from the endemic regions of Africa in the future.

## 2. Materials and Methods

### 2.1. Isolation of Virus Samples

A subset of well-characterized ASFV-positive tissue samples was selected from a study in West and Central Kenya and Eastern Uganda [17]. The samples represented ASFV isolates obtained from different locations in Kenya and Busia district of Uganda in 2012 and 2013. The isolates were selected to represent a diverse spectrum of origins, including samples from outbreaks from multiple geographical locations and dates, long-term infections (isolated through horizontal sampling) and slaughter slabs. All were positive according to diagnostic PCR, but qPCR was not performed, so the viral copy number was unknown. A map of the sampling region is included in Figure 1, and the key metadata for the nine isolates are described in Table 1. The sequences of p72 and other polymorphic loci for these isolates, including the B602L chaperone encoding gene, are available in the INSDC databases. Five of these were sequenced following a single passage on porcine primary macrophages from domestic pigs, as described in [17].

### 2.2. DNA Extraction

DNA was extracted from the infected primary macrophage isolates using the DNeasy Blood and Tissue Kit (Qiagen, Hilden, Germany) in accordance with the manufacturer’s instructions. For the three field samples, DNA was extracted directly from the spleen, kidney and lung tissues from clinically reacting and long-term ASFV-infected pigs in the field in Kenya. DNA extraction also used the Qiagen blood and tissue kit with the modification that presented an elution volume of 30 to 50 µL.

### 2.3. Sequencing Strategies

Five Kenyan and Ugandan viral isolates, passaged once on porcine macrophages [17], were determined using the MiSeq platform. These were the original Genotype IX KE/2013/Busia.3.field, also known by the laboratory codename “1033”, as well as a variant thereof which has been used for live attenuated vaccine research following multiple passages on wild boar cells and experimental attenuation (KE/2013/Busia.3.WSL, published here but already used in [18,19]). Sequencing of this 5× WSL-passaged version used an Illumina MiSeq with MiSeq^®^ Reagent Kit v3 (600 cycles), # MS-102-3003 in 2 × 300 bp paired-end mode. Libraries were prepared with the Beckman Coulter SPRI-TE instrument using SPRIworks Fragment Library Kit II-50: DNA fragmentation was performed using a Covaris Ultrasonicator M220 with a Covaris M220 AFA Ultrasonicator microTUBE AFA Fiber Screw-Cap (6 × 16 mm (250), 130 µL), following automated size selection with the SPRI-TE system and further manual size selection. Other isolates were KE/2013/Busia.8, KE/2013/Kiambu.2, UG/2013/Alupe.1 and KE/2013/Nakuru.1 (see Table 1). For processing of these MiSeq samples, the procedure has already been documented [11]. Briefly, genomic DNA samples were further purified using Agencourt AMPure XP beads (Beckman Coulter Life Sciences, Indianapolis, IN, USA) following the manufacturer’s instructions. Illumina library preparation employed the Nextera XT library preparation kit according to the manufacturer’s protocol. For each sample, tagmentation was performed, and the resulting library was amplified. The amplified Nextera XT library thus generated was purified using Agencourt Ampure XP beads. For whole-genome sequencing, DNA libraries were quantified using the KAPA Library quantification kit for Illumina platforms (KAPA Biosystems, Wilmington, MA, USA). Equal concentrations of the sample libraries (n = 4) were pooled and sequenced as a multiplex on a MiSeq (Illumina Inc., San Diego, CA, USA) machine using the Illumina Reagent Kit version 3 (600 cycles). Raw fastq reads thus obtained were trimmed to remove adapter sequences and low-quality reads. Demultiplexed fastq files were converted to fasta and reads used to generate contigs for downstream analysis.

In order to maximize coverage of the ASFV genome, sequencing of the three DNA samples obtained directly from infected kidney tissues of pigs from the field (KE/2013/Siaya.4, KE/2012/Busia.564 and KE/2013/Karen.1) was outsourced to Genohub Inc. (Austin, TX, USA), where the high-throughput Illumina HiSeq X platform was used with paired 150 bp reads (Illumina Inc., San Diego, CA, USA). The library for whole-genome sequencing was prepared with a KAPA HyperPlus Kit (KAPA Biosystems, Wilmington, MA, USA) using 10 ng of DNA as input according to the manufacturer’s protocol. The final quality and quantity of the library were analyzed with an Agilent Bioanalyzer 2100 and a Life Technologies Qubit 3.0 Fluorometer.

The raw reads generated from the Illumina MiSeq and HiSeq X platforms were further analyzed at the International Livestock Research Institute in Nairobi, Kenya.

### 2.4. ASFV Genome Assembly

The short reads from the Illumina platform were first trimmed for poor-quality ends and adapter content using Trimmomatic [20] version 0.38 in the following fashion. After removing Illumina adapters (commandline filter ILLUMINACLIP:2:30:2 with the appropriate TruSeq3 adapters), we performed a light quality-based trimming only, using a sliding window of length 4 to trim base pairs from the end of the reads as long as the average Q-score in the window remained below 15 (commandline filters LEADING:10 TRAILING:10 SLIDINGWINDOW:4:15 MINLEN:20). Such a light quality-based trimming was performed on purpose, in order to maximize the information content in the input to the assemblers used, which come with their own error-correction algorithms (BayesHammer by default in SPAdes/Unicycler on Illumina reads; see below). We then used Bowtie2 [21] version 2.3.4.1 to map the trimmed reads (both paired and unpaired survivors) to the reference pig genome (Sscrofa11.1, RefSeq assembly accession GCF_000003025.6), complemented with a set of 327 complete pig mitochondrial sequences obtained from the public INSDC database. We excluded the mapped reads from the following steps of the pipeline, thus removing the excess of host reads. Bowtie2’s --unal commandline parameter was used, together with the --no-discordant one for paired reads. We then used SPAdes 3.15.5 [22] or, when the assembly with SPAdes was unsuccessful, Unicycler [23] version 0.4.7 (with SPAdes 3.13.0 as a primary aligner) to perform the assembly of the resulting trimmed, non-pig reads, with the default command line parameters. For seven out of nine samples, we report the longest contig obtained from the assembly as a complete genome, after checking it aligned well to known full-length ASFV sequences. For the two isolates KE/2013/Busia.8 and KE/2013/Nakuru.1, the full-genome construct was obtained by stitching together two contigs at low-depth regions. More specifically, the assembly of KE/2013/Busia.8 yielded two contigs of lengths 116,726 bp and 68,220 bp, which we stitched together with a stretch of 38 Ns after mapping that gap onto an equally long, unambiguous and fully conserved region of both KE/2013/Kiambu.2 and UG/2013/Alupe.1 (Appendix A). The assembly of KE/2013/Nakuru.1 produced two contigs of lengths 169,501 bp and 15,606 bp. These two contigs overlapped around a 9- to 13- long guanine homopolymer repeat, the flanking regions being unambiguously aligned to the last three abovementioned genomes (Appendix A). Whenever necessary, the contigs were reverse-complemented so that our genomes were all published in the same strand as the already published full-length virus sequences. Sequence manipulation and visualization were performed with Unipro UGENE [24]. Our resulting full genomes ranged between 182,715 bp and 187,043 bp.

### 2.5. Assembly QC by Mapping the Short Reads Back

In order to check the quality of the constructs output by the assembler, we mapped the reads used as input to the assembly process (i.e., the initial Illumina reads after trimming with Trimmomatic and after removal of reads mapping to the pig host) back onto our final assemblies. Apart from the missing read data for KE/2013/Busia.3.WSL (the sequencing was performed years ago at the Friedrich Loeffler Institute by a former staff member, and we could not recover the reads), this remapping of the reads for eight of the nine isolates was carried out using the most recent bwa-mem2 version 2.2.1 [25], with all its default parameters. We then used samtools depth with its -a option [26] in order to map to all positions of the assemblies and accurately report the 0-depth scores, if any. The resulting key descriptive statistics for the distribution of the read depths along our assemblies are presented in the last column of Table 1.

### 2.6. Genome Annotation

Recognizing the shortcomings of the previously published annotation for Ken06.Bus (accession number KM111295.1, 184,368 bp, 161 annotated genes) versus the more recently completed Genotype IX sequences such as the five Ugandan sequences determined in [16], we decided to adopt a uniform scheme of annotation transferred from the R25 sequence (accession number MH025918.1, 188,630 bp, 172 annotated genes) as an annotation donor for all our novel Genotype IX sequences. We used RATT’s “Strain.Global” transfer type [27], as well as the default annotation transfer with Prokka’s —proteins option [28].

The eight novel, annotated Genotype IX genome sequences (five isolates from field outbreak sampling and three from spontaneous field surveys, including one sample from a slaughter slab), together with a 5×-passaged wild boar lung cell (WSL) version of the isolate KE/2013/Busia.3 (a.k.a. “1033”) [18,19], were submitted to the European Nucleotide Archive (ENA) and were given the INSDC accession numbers OZ002814, OZ002854, OZ003275, OZ003747, OZ005798, OZ005800, OZ005801, OZ005803 and OZ005804, all under BioProject PRJEB70459.

### 2.7. Phylogenetic Analysis

In order to prepare a phylogenetic tree capturing as much as possible of the full-length variation in ASFV genomes worldwide, showing the accurate phylogenetic placement of our nine novel sequences against the background of all known genomes, we proceeded as follows. We included, in our initial multiple sequence alignment (MSA), all publicly available (as per 30/01/2024) full-length sequences of Genotypes VIII, IX and X. In addition to these, we selected a sample of “historical” ASFV sequences pertaining to Genotype I (such as BA71V, Benin 97/1 and E75), as well as some other more recent Genotype I genomes, including a recent isolate from Cameroon, to bring the number of gen.I sequences to a total of nine. We added 21 Genotype II sequences with a great geographic span, including the recently published Tanzania/Rukwa/2017/1 [10], together with the re-sequenced ASFV Georgia 2007/1 from Georgia [29], Pig/HLJ/2018 from China [30], Belgium/Etalle/wb/2018 [31] and a number of other African isolates. We added four full-length “outlier” sequences from Southern Africa, each one being the single known representative for their respective p72 genotypes: Tengani 62 of Genotype V and of high virulence [32], “Warmbaths” (Genotype III) and Pretoriuskop/96/4 from South Africa (Genotype XX) [33] and the “Warthog” isolate from Namibia, within p72 Genotype IV [33].

With this set of 57 full-genome sequences, we used Mafft version 7.505 [34] to build a multiple sequence alignment (with the default command line options plus—memsave). In order to reduce the amount of noise brought by the gap-rich and saturated sites of the alignment, we used BMGE [35] with its default parameters to trim these sites, bringing the MSA from 227,357 sites down to 171,989 sites. We then used FastTree version 2.1.11 [36] to infer a phylogeny using a GTR-CAT model (General Time-Reversible with rate CATegories) with a discrete gamma model featuring 20 rate categories stretching the branch lengths to fit slow- and fast-evolving sites when computing likelihoods (command line options -nt -gtr -quote -gamma), obtaining a tree that we present here.

## 3. Results

### 3.1. Phylogenomics of ASFV Genomes: p72 Genotypes IX and X Are Highly Divergent from Genotypes I and II

A maximum likelihood phylogenetic tree containing 57 ASFV taxa, representative of all p72 genotypes for which complete genomes are publicly available, was constructed as explained above and is shown in Figure 2. In Appendix A, the reader may find the same tree represented as a cladogram with local statistical supports for the splits calculated as per the Shimodaira–Hasegawa test [37]. The maximum likelihood tree includes the nine new Genotype IX genomes from Kenya (eight genomes) and Eastern Uganda (one genome) determined in this study, plus five previously described Ugandan ASFV Genotype IX genomes from the Tororo region [16], the Kenyan Ken06.Bus isolate from Western Kenya [12] and the recently published TAN/16/Magu sequence from Tanzania [38]. Also included in the tree are five previously determined genomes classified within the closely related p72 Genotype X clade [12,39,40,41]. Additionally, we selected well-annotated representatives from the ASFV Genotypes I and II present in the public databases and single genomes from Genotypes III, IV, V, VIII, XV and XX [39]. Overall, the ML tree topology is similar to that previously determined using 125 concatenated conserved ORFs (Figure 1A in [39]), although the earlier study included only a single full genome of Genotype X (the “Kenya 1950” isolate) from the East African Genotype IX/X clade, and no Genotype IX genome. The tree presented here includes 16 near full-length ASFV Genotype IX genomes (excluding the hairpin ends). All 16 Genotype IX genomes, including 9 sequences determined in the current study, clustered together in a clade with very limited internal diversity but relatively close to the Genotype X clade, and clearly diverging from all other genotypes.

Using the whole-genome multiple sequence alignment of the 57 isolates we included in this study, we went on to create a raw sequence similarity matrix between all 16 sequences of Genotype IX, excluding gaps in a pairwise fashion, and recording the levels of sequence identity (Figure 3). All Genotype IX sequences exhibited very strong percentages of sequence similarity, the lowest level being 99.89%. The first representative of the Genotype IX clade, genome Ken06.Bus of INSDC accession KM111295 [12], isolated from an outbreak in 2006, displayed high sequence similarity with the newly determined genomes, at a whole-genome level of 99.93% on average. Taken together, the genome sequences from p72 Genotypes IX and X are very distinct from all other genomes. Genotypes IX and X, however, comprise phylogenetically distinct groups (Figure 2). However, there is significantly more divergence within the Genotype X clade than within the Genotype IX cluster. Our phylogeny highlighted Genotypes II and IX as being very distant from each other, each one displaying very little internal diversity compared with the broad picture of ASFV worldwide. Such a relatively low level of divergence is probably a sign of high genetic fitness, underlying the fact these two genotypes are “doing well” while meeting little to no evolutionary pressure, each in their own geographic area of prevalence.

### 3.2. The Genome of KE/2013/Busia.3.field, a Kenyan Field Genotype IX Reference Strain, Compared with Its Wild Boar Lung Cell-Passaged Version KE/2013/Busia.3.WSL, Used for Live Attenuated Vaccine Research

Among the set of five Genotype IX p72 genomes determined using the MiSeq platform at ILRI is the original field isolate KE/2013/Busia.3.field [17], which was code-named Isolate 1033. As mentioned, a version of this virus passaged on wild boar lung (WSL) cells at the Friedrich Loeffler Institute (FLI) and incorporating a deletion of the “CD2-like” ASFV locus has shown promise as an experimental live attenuated vaccine [18]. The KE/2013/Busia.3.field virus without any deletion remains virulent in pigs after propagation, and whole-genome sequencing appears to show relative genetic stability following extensive propagation on WSL cells [19].

We compared the original field 1033 isolate with the virus that was passaged 20 times on wild boar lung cells. We detected five differences. The most significant differences involved the terminally located multicopy gene families. MGF360-1L contained an insertion at Position 2149, and there was a SNP in a stop–start codon at Position 2194, resulting in the fusion of MGF360-1L and MGF360-2L in the WSL-passaged version of 1033-IX (KE/2013/Busia.3.WSL, INSDC accession OZ005801). In addition, there was a premature stop codon introduced in a poly-A tract due to a deletion in MGF300-2R resulting in truncation of the predicted protein to 82 aa instead of 160 aa in the field virus. The other two changes involved a synonymous SNP and a mutation in an intergenic poly-G region that probably lacks functional significance.

### 3.3. Genotypes VIII, IX and X Harbour Polymorphisms in the p72 Genotyping Primers

A consistent mismatch in the sequences of p72 Genotypes IX and X relative to other genotypes altered the standard primers p72U and p72D [8] used to amplify the C-terminal region of p72 for genotype determination. p72U is a 19-nucleotide sequence readable directly on the bottom strand, with the sequence 5′ GG/CAC/AAG/TTC/GGA/CAT/GT 3′ starting at Position 97,875 of the reference full-genome sequence of the E75 strain (since it is within the B646L coding region, the codons’ boundaries are indicated with slashes). In the Malawi_Lil-20/1 reference strain (of Genotype VIII), the third nucleotide from the 3′ end of the p72U sequence is a C instead of a T (Position 102,422 of the Malawi_Lil-20/1 sequence, corresponding to 97,859 in E75). The polymorphism is synonymous, since both CAT and CAC encode for histidine (H).

The 19-nucleotide p72D primer (e.g., starting at Position 97,398 of E75) reads 5′ GT/ACT/GTA/ACG/CAG/CAC/AG 3′ on the top strand and, therefore, is complementary to the sequence of the B646L CDS. Here again, we indicated the codons’ boundaries with slashes. At the third position of p72D, four of the five full-genome sequences of Genotype X (Kenya 1950, Uvira B53, BUR/18/Rutana and Ken.rie1) exhibited a polymorphism, with a G instead of an A (a synonymous SNP, since both AGT and AGC code for a serine). Further into p72D, at Position 14 of the primer, there was a G/A SNP (at Position 97,411 in E75 and Position 100,990 in the KE/2013/Busia.3.WSL full-genome sequence) that was common to all Genotype IX and X sequences, including all nine newly sequenced Genotype IX ASFVs presented here. Although in the first codon position on the coding strand, it is synonymous, since both the codons CTG and TTG code for a leucine (L). Despite the fact that the pair p72U/p72D has been used successfully for decades in the detection of African swine fever virus, including viruses of Genotypes IX and X [17,44], it is not completely out of question that such a polymorphism towards the end of a primer may decrease the efficiency of the annealing and might lead to detection errors in diagnostic assays based on the amplification of that region. In Figure 4, we represent the multiple sequence alignments of the end of the B646L CDS for all Genotypes IX and X in this study, contrasting with the other sequences and clearly displaying the two polymorphisms described above in the reverse complement of p72D.

### 3.4. Conserved p72 Genotyping Sequences among All Genotype IX Genomes

The 16 sequences of Genotype IX were all strictly identical in the portion of the B646L region amplified by the p72-U and p72-D primers (except that TAN/16/Magu was annotated with a shorter ORF). The whole B646L gene, which encodes for the capsid protein, displayed only three SNPs outside of the primer-amplified region:A synonymous G/A SNP at Position 75 of the coding sequence (the third codon position for a leucine), for which all nine novel Genotype IX sequences had an A, whereas all previously published Genotype IX sequences contained a G;An isolated synonymous C/T SNP at Position 84 (the third codon position for a serine) appeared only in KE/2013/Karen.1;A C/T SNP at Position 1047 (the third codon position for a serine) had all five isolates from [15] with a T, and all others with a C.

### 3.5. Central Variable Region (CVR) Sequences in the B602L Gene Revealed New Amino Acid Tetramers in Genotype IX Genomes

Since the seminal paper by Nix and co-authors [45], the so-called central variable region, located within the B602L gene, has been highlighted by many studies as showing great diversity, including within a single country such as Tanzania [39,46,47,48]. The CVR encodes a well-described stretch of tetrameric amino acid repeats that are known to evolve at a very rapid pace, representing the only locus that is useful for the discrimination of variants within an outbreak [45]. Inspection of the CVR sequences pertaining to the 16 isolates of Genotype IX involved in this study revealed the following results (see Table 2), which contained a few minor discrepancies in comparison with earlier reports in [16].

We confirmed a new tetramer that was only hinted at in [17], CADI. It was a consistent occurrence in all ten of “our” Genotype IX genomes, i.e., the nine novel ones plus Ken06.Bus [12]. It aligns with the CADT (nicknamed B) tetramer shown here by the Ugandan isolates from [16]. We labeled this CADI tetramer with the letter E.TAN/16/Magu alone [38] had a novel tetramer CVDI, substituting for the abovementioned CADI. Since valine and alanine are not very distant in terms of their physicochemical properties, this is likely to represent a true aa polymorphism. However, since it is a unique tetramer never reported previously by authors studying the CVR region, it should ideally be confirmed by the authors of [38]. If validated, the data would provide additional evidence that CADI should not be lumped with CADT under the same B category (as described in [17]) but is in itself a “core” tetramer that is able to mutate into the derived form, CVDI.All Ugandan Genotype IX genomes described in [16] started their tetrameric repeat region with four As (representing four CAST tetramers), while all other Genotype IX isolates (including our novel nine isolates plus Ken06.Bus and TAN/16/Magu) started with only three As.Five of our nine novel genomes introduced the substitution of a CVST tetramer (coded a, as a variant of A, which is CAST) with a CADT tetramer (B).Ken06.Bus had only one CADT (B) in a locus, whereas all other Genotype IX viruses had two copies of this tetramer (BB).

All these results are summarized in Table 2. They illustrate the high level of plasticity of the B602L CVR sequences found within Genotype IX viruses, as well as the high level of divergence of Genotypes IX compared with, e.g., the well-studied Genotypes I and II.

## 4. Discussion

The current study has significantly increased the pool of complete genome sequences available from Genotype IX and confirmed that, together with Genotype X, these East African viruses differed significantly in genome sequence from other genotypes but were relatively conserved in populations from within the region. This is the first comprehensive dataset that allows an examination of the population genomics of ASFV from a distinct region within Sub-Saharan Africa, which is the ancestral home of the virus. In the geographical areas that we sampled, from Eastern Uganda to Central Kenya and Northern Tanzania (for Genotype IX) and in Kenya, Burundi and Eastern DRC (for Genotype X), the genomes were closely related in the core set of conserved ASFV genes [49]. There was no evidence for more than one p72 genotype virus being present in any of these isolates, although targeted deep sequencing of the p72 locus using samples from multiple tissues from individual domestic pigs would be required for formal confirmation that ASFV co-infections do not occur in the field.

Differences between the individual ASFV isolates typically involve inserts and deletions (indels), which are primarily located within the N-terminal and C-terminal multicopy gene families, MGF 100, 110, 360 and 550/530 [49,50,51]. These may be the result of ectopic recombination involving unequal crossovers between non-homologous sequences in multicopy gene families [52]. This has led to the perception that the ASFV genome is exceptionally labile for a large DNA virus, and differences have been observed between outbreak viruses within Georgia and China. These observations suggest rapid evolution of the Genotype II virus under epidemic conditions since its introduction to novel pig-food epidemiological environments in Eurasia in 2007. In the areas of East Africa that we analyzed, there is only a single virus genotype circulating, although Genotypes IX and X both appear to be present in Eastern DRC, with Genotype X viruses apparently the major cause of the disease.

The indels represent single mutational events and potentially distort the more conservative picture of population genomics that we describe within two endemic regions of Africa. There are currently variant genomes described even within Georgia, where the Genotype II virus was originally introduced to Eurasia from Southeastern Africa [53] and also China [54]. However, there is no evidence for co-infection of different genotypes within the endemic area of Africa that we studied, suggesting that classical homologous recombination may currently be rare in ASFV among domestic pigs in the indigenous areas of Africa, although it possibly occurs more frequently in the warthog–tick sylvatic cycle, or in specific areas of the genome such as the adjacent CD2v (EP402R) and C-type lectin (EP153R) loci, which display great variability even within a given genotype such as Genotype X [55,56].

Despite the relative similarity of the genomes of p72 Genotypes IX and X, it is interesting to note that the isolates within Genotype X were more closely related to one another at the whole-genome level than they were to any of the 16 Genotype IX genomes. It therefore appears that although their p72 sequences are relatively similar, Genotypes IX and X represent different clades that have presumably separated relatively recently but appear to be clearly distinct. Indeed, the evolutionary distance at whole-genome level between genomes from Genotypes IX and X was actually slightly larger than the distance between Genotypes I and II. For instance, the phylogeny we present in Figure 2 showed an evolutionary distance (expressed as the expected number of mutations per aligned position) of 0.039 between our nine isolates and the historical Kenya 1950 Genotype X isolate, while the evolutionary distance between Georgia 2007/1 (Gen. II) and Benin 97/1 (Gen. I) was 0.034.

A recent study based on full-length, rather than partial, p72 sequences identified only six distinct clusters, in which Genotypes IX and X were classified together [42]. However, the use of an ML-based algorithm derived from 220 open reading frames identified through a comparative analysis of whole-genome sequences of ASFV clearly separated Genotypes IX and X into different biotypes, numbered, respectively, 5 and 4 [43]. The latter finding is consistent with the whole-genome analyses of Genotypes IX and X from the current study, in contrast with the poor discrimination power obtained when focusing only on full-length p72 sequences [42].

Recent literature suggested that Genotype X viruses are primarily responsible for clinical outbreaks in pigs from the Eastern DRC [57], whereas the recent isolates of Genotype X from East Africa were both from *Ornithodoros* ticks in warthog burrows at a single locality in Kenya. Although they originated from the same ranch, these two Kenyan tick-derived isolates were relatively distinct at the genome level, highlighting the observation that ASFV isolates from ticks are more variable than those from pigs [1].

The Malawi Lil-20/1 isolate, classified within p72 Genotype VIII, which primarily contains genotypes of Zambian origin, is an outlier in the ML whole-genome tree, as also observed in earlier analyses [39]. It is now joined by TAN/08/Mazimbu of Genotype XV. Genome sequences from southern and southern-central Africa, West Africa and Europe showed reasonably low divergence at the whole-genome level, with one clade for the West African and European Genotype I isolates, another tight clade for the Genotype II sequences and a third cluster representing multiple p72 genotypes, namely III, IV, V and XX. This indicated that although the p72-gene-based classification remains important as an initial classification tool, the extra data provided by whole-genome sequencing adds significant value to phylogenetic analyses of ASFV.

To date, less than 50% of the currently described p72 genotypes have a corresponding whole genome, and before this study, only Genotypes I and II had eight or more full genomes, and these are direct descendants of a small number of virus “migrations” out of the African continent. The improved rapid genome sequencing protocol described herein, using the Illumina Hi-Seq-X platform applied to a template of total mammalian host and viral DNA, extracted directly from field tissues (spleen, kidney or lung) sampled on ASFV-infected animals, or from primary cultures, will hopefully enable the rapid completion of genomes from additional p72 genotypes present in the endemic areas of Africa. This will allow more comprehensive analyses of the virus’s evolution, population genomics and phylogeography in the future.

## Figures and Tables

**Figure 1 viruses-16-01466-f001:**
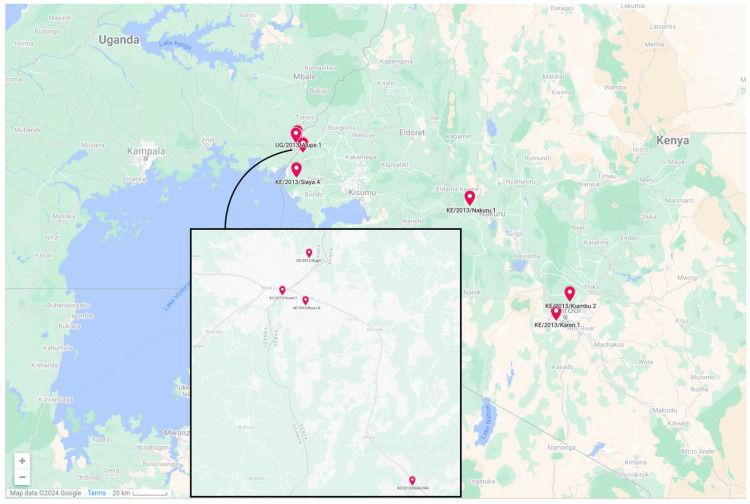
Geographical map of the sampling sites in Kenya and Uganda.

**Figure 2 viruses-16-01466-f002:**
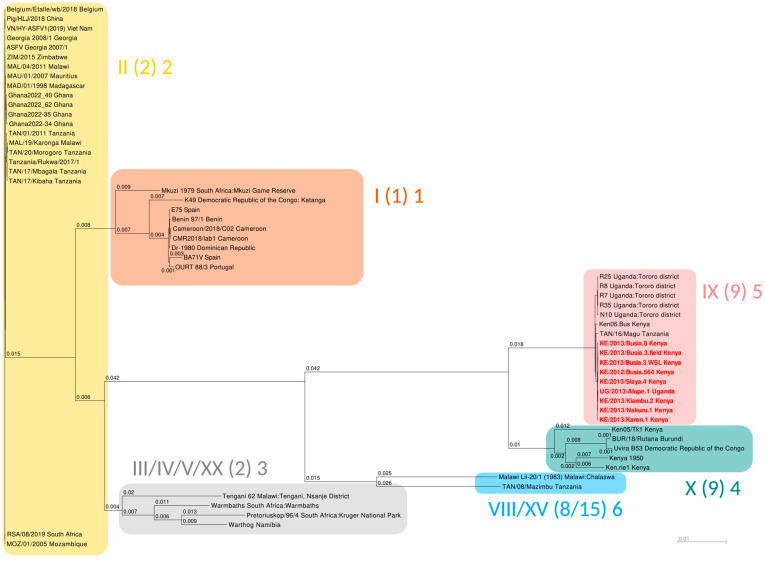
Maximum likelihood phylogenetic tree of ASFV isolates based on a whole-genome multiple alignment. Colored clades are highlighted according to: (i) historical p72 genotypes based on the 3′ end of the B646L ORF (Roman numerals); (ii) genotype groups as described in [42], derived from full-length p72 protein sequences (parenthesized numbers); (iii) biotypes as described in [43], from a full-proteome, ML-based analysis (Arabic numerals). The branch length values (scale at the bottom right of the figure) represent the mathematical expectation of the number of nucleotide substitution events in the pairwise alignment between the two sequences (ancestral or extant) present at the tips of any given branch. Branch length values lower than 0.5 × 10^−3^ are not displayed. This figure was prepared with iTOL (https://itol.embl.de/) before post-processing in Inkscape.

**Figure 3 viruses-16-01466-f003:**
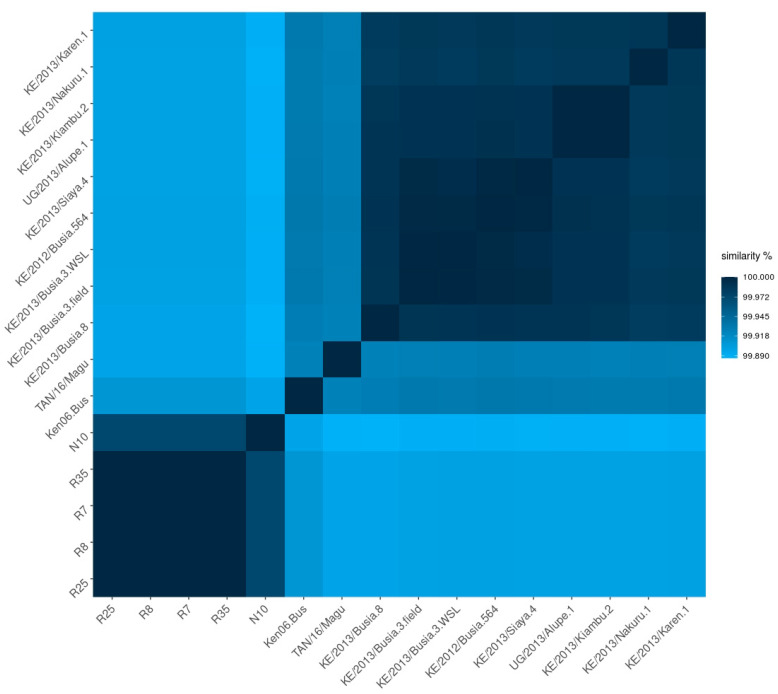
Heatmap of whole-genome sequence similarity across all available Genotype IX genomes.

**Figure 4 viruses-16-01466-f004:**
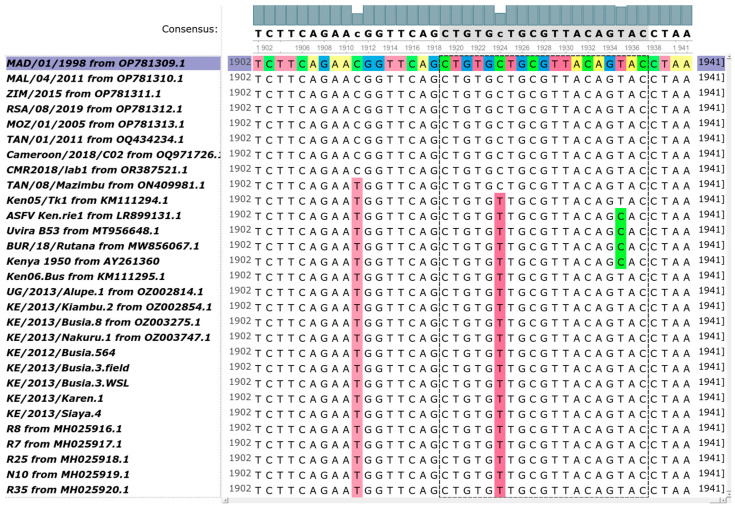
The 3′ end of the coding sequences for B646L/p72 in a multiple sequence alignment including all published full-length genomes of Genotypes IX and X, among which are the nine Genotype IX genomes from this study. The 19 bp region framed with dashes corresponds to the complementary region for p72D. It harbors the two SNPs described in the text, the first of which (at Site 1924) is common to all viruses from Genotypes IX and X. The topmost, coloured sequence is a genotype II sequence chosen as a random reference so that the SNPs below are then highlithed with respect to that sequence. This image was prepared with UGENE [24].

**Table 1 viruses-16-01466-t001:** Synthetic passport data for the nine new viruses published here. * In isolate KE/2013/Busia.8, the read depth from mapping the reads back to the assembly logically fell down to 0 within the filler of 38 Ns used to perform the stitching of two contigs, and was 1 only in the immediate vicinity of the stitch, within bp positions 68,182 and 68,384 of the published sequence (see main text). ** In isolates KE/2013/Busia.8 and KE/2013/Nakuru.1, the read depth from mapping the reads back to the assembly fell to 1 on the single final base of the assembly at bp positions 184,984 and 185,088, respectively.

Isolate Name	INSDC Accession	Previously Published Sample Alias, Internal Sample Identifiers	p72 Genotype	Location and Specific Origin	Rounded GPS Coord. (Decimal Degrees)	Sampling Date (YYYY/MM/DD or YYYY/MM)	Isolation Tissue	Published Genome Length (bp)	Final Mean k-mer Coverage (SPAdes) or Mean Depth (Unicycler)	Min, Median and Max Read Depth upon Mapping Reads Back
KE/2013/Kiambu.2	OZ002854	ken13/kiambu.2, p36/13, p36-13_S4	IX	Kiambu County, Kenya: outbreak; tissue supplied by the Directorate of Veterinary Services (DVS)	−1.00, 36.80	2013/08	Kidney	187,043	71.7	10, 101, 401
UG/2013/Alupe.1	OZ002814	ug13/alupe.1, Uganda 231, Ug-231_S5	IX	Alupe, Uganda. Adjacent to the Kenyan border: longitudinal sample sequestered in pig tissue	0.49, 34.11	2013/02	Spleen	186,980	55.2	5, 76, 358
KE/2013/Busia.8	OZ003275	ken13/busia.8, Odukin 1	IX	Busia, Western Kenya: slaughter slab	0.56, 34.27	2013/10	Lung	184,998	20	2 *^,^**, 26, 246
KE/2013/Busia.3.field	OZ005798	ken13/busia.3, Busia-Teso 1033, Kenya_IX_1033	IX	Busia–Teso boundary, western Kenya: outbreak sampling	0.46, 34.10	2013/02	Spleen	182,717	85.08	27, 233, 895
KE/2013/Busia.3.WSL	OZ005801	ken13/busia.3, Busia-Teso 1033, Kenya_IX_1033	IX	Busia–Teso boundary, western Kenya: outbreak sampling	0.46, 34.10	2013/02	Spleen	187,015	N/A	N/A
KE/2013/Siaya.4	OZ005800	Nyadorera 1	IX	Nyadorera, Siaya County, western Kenya: outbreak sampling	0.11, 34.10	2013/09/13	Spleen	182,882	33.4	24, 82, 2732
KE/2012/Busia.564	OZ005803	2540564	IX	Matayos, Busia County, western Kenya. Pig sampled during a field survey (cross-sectional sampling)	0.36, 34.17	2012/09/28	Spleen	182,715	354.15	211, 353, 7889
KE/2013/Nakuru.1	OZ003747	ke13/nakuru.1, Rongai 1, Rongai_S1	IX	Rongai, Nakuru County, Kenya: outbreak; tissue supplied by the DVS	−0.17, 35.86	2013/08	Kidney	184,963	43.6	2 **, 58, 327
KE/2013/Karen.1	OZ005804	P38	IX	Karen, Nairobi county, Kenya: suspected outbreak; tissue supplied by the DVS	−1.30, 36.67	2013/09/01	Spleen	182,720	151.36	20, 153, 7368

**Table 2 viruses-16-01466-t002:** Tetrameric signatures of the CVR (B602L) region for 16 Genotype IX viruses. Polymorphic loci are highlighted. Amino acid tetramers are coded as follows. A, CAST; a, CVST; B, CADT; E, CADI; e, CVDI; N, NVDT.

Aligned Tetrameric Sequences	Isolates
AAA**A**BNA**B**BNABB**aaBB**NABNaBA	R7, R8, R25, N10, R35 [16]
AAA**-**BNA**E**BNABB**aaB-**NABNaBA	Ken06.Bus [12]
AAA**-**BNA**e**BNABB**aaBB**NABNaBA	TAN/16/Magu [38]
AAA**-**BNA**E**BNABB**aBBB**NABNaBA	KE/2012/Busia.564, KE/2013/Busia.3.field, KE/2013/Busia.3.WSL, KE/2013/Busia.8, KE/2013/Siaya.4 (this study)
AAA**-**BNA**E**BNABB**aaBB**NABNaBA	UG/2013/Alupe.1, KE/2013/Kiambu.2, KE/2013/Nakuru.1, KE/2013/Karen.1 (this study)

## Data Availability

All data pertaining to this research are available in the public INSDC databases under BioProject PRJEB70459. The data include BioSamples SAMEA115043951 to SAMEA115043959, as well as the nine corresponding annotated genomes (and the derived protein sequences) with the INSDC accession numbers OZ002814, OZ002854, OZ003275, OZ003747, OZ005798, OZ005800, OZ005801, OZ005803 and OZ005804.

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
