# Peer review of "Complete Genome Sequencing and Comparative Phylogenomics of Nine African Swine Fever Virus (ASFV) Isolates of the Virulent East African p72 Genotype IX without Viral Sequence Enrichment"

_viruses, 2024, doi:10.3390/v16091466_

Round 1
Reviewer 1 Report (New Reviewer)
Comments and Suggestions for Authors
The manuscript presents novel data relevant for understanding epidemiology of African swine fever virus in Africa. The genotype IX virus sequences presented also provide templates for development of vaccines for regions of East Africa where genotype IX is circulating. The data is clearly presented and described.
Author Response
Comments 1: The manuscript presents novel data relevant for understanding epidemiology of African swine fever virus in Africa. The genotype IX virus sequences presented also provide templates for development of vaccines for regions of East Africa where genotype IX is circulating. The data is clearly presented and described.
Response 1: We thank the reviewer for this very positive feedback which doesn't call for any response. We are glad we presented convincing data and results that are fit for publication.
Reviewer 2 Report (New Reviewer)
Comments and Suggestions for Authors
Domelevo Entfellner et al. expand our knowledge about the genetic diversity of African swine fever virus in Eastern Africa by generating genome sequence data for 9 new isolates from Kenya and Uganda. The data the authors present generally supports their conclusions, my main concerns are methodological and I believe the authors have over interpreted the significance of the finding of a SNP in a common genotyping primer.
Main points.
Line 186. Section 2.4. Please indicate the quality threshold and length of the retained reads in the text for those who are not familiar with Trimmomatic. My interpretation of the command line parameters is that a minimum quality threshold of 10 was used, rather than the more standard 30. Please indicate why low quality reads were used. Were reads mapped back to the final assemblies to check for errors? This is step is essential and the minimum coverage for each genome should be included in Table 1, ideally this should be done with high quality reads.
Minor points
Line 57. One of the p72 genotypes has recently been retired (see https://doi.org/10.1128/mra.00067-24), suggest removing references to the precise number of genotypes here and at Line 515.
Line 256. What was model was used for the maximum likelihood trees? Was any bootstrapping performed to confirm support for the tree structure?
Figure 2. Please indicate in the legend what the numbers on the branches on Figure 2 are. In addition, please separate these numbers from the names of the virus isolates as they are overlapping in some cases.
Line 350. Section 3.3. The conclusions of this section aren’t backed up with evidence and are a little speculative. The p72-U and p72-D primers have been used successfully on every ASFV isolate that has been characterized for the last two decades, including the viruses that are the focus of this paper (see reference 17), more than 30 other viruses of genotype X (https://doi.org/10.1099/vir.0.025874-0) and a similar number of genotype VIII viruses (https://doi.org/10.1007/s11262-007-0148-2). Please include evidence that these differences effect the genotyping assay or paraphrase the section and move to the discussion.
Line 422. This sentence should be qualified to state that genotype IX has high divergence within the B602L CVR sequence. As the authors have demonstrated at the genome level they are relatively homogeneous.
Line 459 to 466. This section on recombination should include a brief description of the work from the Pokrov lab as there are significant differences in the EP402R gene between different genotype X viruses (https://doi.org/10.1099/jgv.0.000024).
Line 468 to Line 478. The references the authors cite here don't support their conclusions. Reference 51 used real time PCR to detect ASFV genome and reference 52 used endpoint primer sets PPA1/PPA2, which are different to P72-U and P72-D primers in which the authors describe SNPs within their results sections. Suggest removing this section.
Author Response
Please see the attachment. The new supplementary figure containing the cladogram version of Figure 2 is Supplementary Figure 3, not Supplementary Figure 1 as indicated in my earlier submission of a response to Reviewer 2.

This manuscript is a resubmission of an earlier submission. The following is a list of the peer review reports and author responses from that submission.
Round 1
Reviewer 1 Report
Comments and Suggestions for Authors
In the manuscript, Complete genome sequencing and comparative phylogenomics of nine African swine fever virus (ASFV) field isolates of the virulent East African p72 genotype IX, from field samples without viral sequence enrichment, the authors sequence some ASFV isolates from genotype IX.
Recently there have been many new genomes of ASFV that have been full length sequenced, the sequences in this manuscript are over 10 years old, and have limited value to the current circulating strains of ASFV. However, from a historical perspective new sequences of ASFV have some value, however these seem to be close matches to what was already published, so the novelity of this work is questionable for a journal of this impact.
1. I am concerned that only illumina reads were used to do what appears to be de novo sequencing of ASFV, which is very difficult to do accurately. Here is looks like at least in some situations, that the genomes were aligned to a reference or low coverage areas were stitched together and then aligned to a reference. There have been multiple instances of insertions or changes in orientation for large sequences of ASFV. To confirm if these genomes are accurate, long -read sequencing should be performed ( nanopore/pac bio).
2. My second concern is that the authors show that genotype IX and X are very distinct from genotype I and II. But they don’t take into account some of the new work, that suggests that IX and X are the same genotype by full length P72 comparison, the authors should confirm or use full length P72 for genotyping ASFV. There is also a new report on Biotyping using whole genome comparison, it would be interesting to compare the Biotyping results on these genomes.
3. The authors use CVR as a type of additional classification, this has limited usage in the classification of ASFV, and it should be noted as such, in addition the depth or quality of reads of this region should be reported to determine the accuracy of these nucleotide changes.
Comments on the Quality of English Language
minor edits are needed
Reviewer 2 Report
Comments and Suggestions for Authors
Comments and Suggestions for Authors
This manuscript describes whole genome sequence analysis of eight ASFV field genomes and one lab-adapted isolate belonged to historic p72 genotypes IX from East Africa. The findings are important and useful for those work on ASF and I recommend accepting this manuscript for publication after the following issues are addressed.
Major Issues
Line 27 and 28: Nine novel ASFV genotype IX genomes were generated from DNA isolated 27 directly from field-sampled tissues. This sentence is bit misleading because five of the viruses sequenced were passaged in primary cells and only three samples from kidney tissues were sequenced directly. One of the isolates was also a lab-adapted strain
Line 93 and 94: Please explain on what basis the samples were selected for whole genome sequencing? Was real-time PCR also done on these samples to get an idea of genome copies in the sample ?
Line 148 : Why these three samples were not isolated and used directly in whole genome sequencing?
Line 169 : --- Missing information?
Line 170 : --- Missing information?
Line 188 ---- Missing information?
Page 9 - Phylogenetic tree is not readable. Please upload the file again
Discussion: A new nomenclature for ASF based on whole genome sequences diving ASFV into 7 biotypes has been proposed.
Dinhobl M, Spinard E, Tesler N, Birtley H, Signore A, Ambagala A, Masembe C, Borca MV, Gladue DP. Reclassification of ASFV into 7 Biotypes Using Unsupervised Machine Learning. Viruses. 2023 Dec 30;16(1):67.
Based on the proposed biotypes, historical genotype IX genomes and X genomes have been assigned to biotypes 4 and 5. Referencing the proposed biotypes and discussing the finding from this study in relation to biotypes is recommended.
Minor Issues
Typos
Line 410 – the word “genes” repeated twice
Line 450 – delete “ but appears distinct” and start the next sentence “Recent”